# Osteosarcoma-enriched transcripts paradoxically generate osteosarcoma-suppressing extracellular proteins

Kexin Li[1,2], Qingji Huo[1,2], Nathan H Dimmitt[2], Guofan Qu[3], Junjie Bao[3], Pankita H Pandya[4,5], M Reza Saadatzadeh[4,5], Khadijeh Bijangi-Vishehsaraei[6], Melissa A Kacena[4,7,8], Karen E Pollok[4,5], Chien-Chi Lin[2,4], Bai-Yan Li[1]*, Hiroki Yokota[2,4,8]*

[1]Department of Pharmacology, School of Pharmacy, Harbin Medical University, Harbin, China; [2]Department of Biomedical Engineering, Indiana University Purdue University Indianapolis, Indianapolis, United States; [3]Department of Orthopedic Surgery, Harbin Medical University Cancer Hospital, Harbin, China; [4]Indiana University Simon Comprehensive Cancer Center, Indiana University School of Medicine, Indianapolis, United States; [5]Department of Pediatrics, Indiana University School of Medicine, Indianapolis, United States; [6]Department of Pediatric Hematology and Oncology, Indiana University School of Medicine, Indianapolis, United States; [7]Department of Orthopaedic Surgery, Indiana University School of Medicine, Indianapolis, United States; [8]Indiana Center for Musculoskeletal Health, Indiana University School of Medicine, Indianapolis, United States

*For correspondence:
liby@ems.hrbmu.edu.cn (B-YanL);
hyokota@iupui.edu (HY)

**Abstract** Osteosarcoma (OS) is the common primary bone cancer that affects mostly children and young adults. To augment the standard-of-care chemotherapy, we examined the possibility of protein-based therapy using mesenchymal stem cells (MSCs)-derived proteomes and OS-elevated proteins. While a conditioned medium (CM), collected from MSCs, did not present tumor-suppressing ability, the activation of PKA converted MSCs into induced tumor-suppressing cells (iTSCs). In a mouse model, the direct and hydrogel-assisted administration of CM inhibited tumor-induced bone destruction, and its effect was additive with cisplatin. CM was enriched with proteins such as calreticulin, which acted as an extracellular tumor suppressor by interacting with CD47. Notably, the level of *CALR* transcripts was elevated in OS tissues, together with other tumor-suppressing proteins, including histone H4, and PCOLCE. PCOLCE acted as an extracellular tumor-suppressing protein by interacting with amyloid precursor protein, a prognostic OS marker with poor survival. The results supported the possibility of employing a paradoxical strategy of utilizing OS transcriptomes for the treatment of OS.

## Editor's evaluation

There are no known effective treatments available to date for the treatment of osteosarcomas, the earliest identified bone cancer that can spread to other tissues. In this study, the authors have used novel approaches to identify calreticulin and procollagen C-endopeptidase enhancer (PCOLCE) as osteosarcoma tumor suppressor proteins that inhibit osteosarcoma growth both in animal and in vitro cell culture models. These important findings may provide a basis for the future development of more efficient targeted therapies for the treatment of osteosarcomas.

## Introduction

A sarcoma is a malignant tumor arising from mesenchymal-originated connective tissues (*Vailas et al., 2019*). Osteosarcoma (OS) is the most prevalent form of bone sarcoma, usually occurring in the lower limb of teenagers and young adults (*Taran et al., 2017*). The primary therapeutic regimen is surgery combined with adjuvant chemotherapy. While the standard-of-care MAP therapy using a combination of methotrexate (MTX), Adriamycin (aka doxorubicin [DOX]), and cisplatin (CIS) has significantly improved the survival rate, the metastatic or recurrent OS remains difficult to treat (*Yu et al., 2019*). To explore targeted therapies, prognostic markers in Wnt, PI3K, RANKL, and Notch pathways have been searched in OS pathobiology. However, few targetable mutations have been identified and the efficacy of immunotherapy is controversial (*Wu et al., 2020*). Here, we examined the possible conversion of mesenchymal stem cells (MSCs) into induced tumor-suppressing cells (iTSCs). Paradoxically, the generation of iTSCs requires the activation of tumorigenic signaling. For instance, we have reported that the overexpression of β-catenin in canonical Wnt signaling and the activation of PI3K/Akt signaling can produce iTSCs and tumor-suppressive conditioned medium (CM) (*Liu et al., 2021a*, *Sun et al., 2021*). To develop a novel option for OS treatment, we examined the generation of MSC-driven iTSCs by activating protein kinase A (PKA) signaling.

T lymphocytes have been frequently employed for immunotherapy because of their immunomodulatory properties as well as their availability in the peripheral blood of patients (*Waldman et al., 2020*). Here, we employed bone marrow-derived MSCs and occasionally adipose tissue-derived MSCs since MSCs are implantable stem cells that generate osteoblasts and osteocytes. In this study, the activation of three signaling pathways was evaluated, including Wnt, PI3K, and PKA pathways. PKA activation in MSCs was selected since it generated stronger anti-tumor effects than the activation of Wnt or PI3K. PKA is known as a cAMP-dependent protein kinase, which is triggered by cAMP (*Walsh and Van Patten, 1994*). Using a mouse model of OS in the proximal tibia, we examined the efficacy of CM with and without the administration of CIS. CM was given as a daily intravenous injection or a weekly hydrogel-based administration (*Lin et al., 2022*).

Previous iTSC studies have revealed that the activation of tumorigenic signaling elevates the levels of extracellular tumor-suppressing proteins, including calreticulin (CALR), enolase 1 (ENO1), heat shock protein 90ab1 (HSP), moesin (MSN), and ubiquitin C (UBC) (*Sun et al., 2021*; *Liu et al., 2021a*, *Liu et al., 2021b*, *Sun et al., 2022b*, *Li et al., 2022b*). Importantly, the listed proteins are reported as tumorigenic in many types of cancers, and their pro-tumor actions inside the cells are reversed to the anti-tumor actions in the extracellular domain. To our surprise, the transcript levels of these tumor-suppressing proteins are significantly elevated in sarcoma tissues in the TCGA database. This observation led us to address an intriguing question as to whether the products of transcripts, which are highly expressed in OS tissues, may act as tumor-suppressing proteins in the extracellular domain.

As for the anti-tumor regulatory mechanism, we focused on CALR and procollagen C-endopeptidase enhancer (PCOLCE). CALR was enriched in iTSC CM as a chaperone protein in the endoplasmic reticulum, but it can be located in the cell surface and extracellular matrix (*Gold et al., 2010*). PCOLCE, an enzyme that cleaves type I procollagen (*Steiglitz et al., 2006*), was selected because of the elevated level of its transcripts in OS tissues. We have shown that CALR interacts with CD47 (*Li et al., 2022a*), an integrin-associated transmembrane protein, which stimulates the escape of cancer cells from immune surveillance (*Sick et al., 2012*). Interestingly, both CALR and PCOLCE are reported to interact with amyloid precursor protein (APP) in IntAct molecular interaction database (*Orchard et al., 2014*). According to the TCGA database, the elevated level of APP transcripts of patients with sarcoma is a poor prognostic marker. Collectively, this study demonstrates a novel role of the CALR/PCOLCE-APP axis in OS progression and suggests APP as an OS-specific druggable target.

## Results

### Suppression of the proliferation, migration, and invasion of OS cells by MSC CM

We first evaluated the efficacy of three pharmacological agents for inducing the tumor-suppressing capability of bone marrow-derived MSC-driven CM. Three agents were BML284, YS49, and CW008, which were known activators of Wnt, PI3K, and PKA signaling pathways, respectively. The MTT-based cell viability of three human OS cell lines (MG63, U2OS, TT2 PDX) was not altered by the control MSC

CM without any agent treatments (*Figure 1A*). However, BML284 (BML), YS49, and CW008 (CW) converted CM into the tumor-suppressing agent. Except for MG63 cells with BML284-treated MSC CM, all CM reduced MTT-based viability in three OS cell lines (*Figure 1A*). Among them, the strongest MTT inhibitory effect was obtained with CW008-treated MSC CM, which also decreased scratch-based motility and transwell invasion of TT2 PDX OS cells (*Figure 1B and C*).

Besides bone marrow-derived MSCs, adipose-derived MSCs (aMSCs) were also used to generate tumor-suppressing CM by the treatment with CW008. CW008-treated aMSC CM inhibited the viability, migration, and invasion of TT2 OS cells (*Figure 1D–F*). The tumor-suppressing CM was also generated by cAMP analog, an activator of PKA signaling, while a PKA inhibitor, H89, oppositely generated tumor-promoting CM (*Figure 1—figure supplement 1*). Collectively, the results revealed that the treatment of MSCs and aMSCs with CW008 generated anti-OS CM. Hereafter, this study focused on CW008-treated bone marrow-derived MSC CM (CW-MSC CM) and examined its anti-tumor role and regulatory mechanism using in vitro and in vivo models.

## Compatibility of CW-MSC CM with chemotherapeutic agents

The standard-of-care chemotherapy for OS employs a combination of MTX, DOX, and CIS. One of the major issues for the clinical application is the compatibility of CM with the existing chemotherapeutic agents. Importantly, the simultaneous application of CW-MSC CM significantly lowered the effective concentrations of MTX, DOX, and CIS in both TT2 and U2OS cells (*Figure 2A and B*). Western blot analysis revealed that CW-MSC CM induced the reduction of p-Src and Snail, as well as the elevation of cleaved caspase 3, a marker for apoptotic death, in TT2 OS cells (*Figure 2C*).

Prior to focusing on proteomes in CM in this study, the potential role of nucleic acids and exosomes in tumor-suppressive capability was evaluated. Accordingly, the anti-tumor action was not significantly altered either by DNA/RNA digestion with nucleases or exosome removal with ultracentrifugation (*Figure 2D and E*). Also, size-fractionated CM indicated that tumor-suppressing proteins were distributed in all fractions including proteins above and below 100 kD (*Figure 2F*). Of note, protein size in kD in the horizontal axis indicates the cutoff size, and the bar for 100 kD, for instance, corresponded to the fraction for proteins 100 kD and larger.

## Bone protection by CW-MSC CM with and without CIS

In the NSG mouse model of OS in the proximal tibia, the weekly intraperitoneal injection of CIS (10 μg/kg), the daily intravenous injection of CW-MSC CM, and their combination significantly reduced the degradation of trabecular bone by elevating the bone volume ratio (BV/TV) and bone mineral density (BMD) in 18 days after the inoculation of TT2 OS cells ($2.5×10^5$ cells) (N=6, *Figure 3A*; N=4, *Figure 3—figure supplement 1*). Consistently, the cortical bone in the proximal tibia was also protected by the administration of CW-MSC CM with and without CIS, although the application of CIS alone at a dose of 10 μg/kg did not significantly restore BMD (N=6, *Figure 3B*). We also applied CW-MSC CM weekly using a hydrogel-based delivery system and observed that the weekly administration of hydrogel-embedded CM effectively protected trabecular and cortical bone for OS-induced degradation (N=5, *Figure 3C and D*; N=3, *Figure 3—figure supplement 2*). Notably, compared to the placebo group, immunohistochemical analysis showed that the administration of CW-MSC CM decreased Ki-67 and increased cleaved caspase 3 in tumor-invaded bone sections (*Figure 3—figure supplement 3*).

## Tumor-suppressing proteins in CW-MSC CM

Previous studies for iTSCs showed that MSC CM, generated by the activation of Wnt and PI3K signaling pathways, were enriched with extracellular tumor-suppressing proteins such as CALR, ENO1, HSP, MSN, and UBC. Notably, western blot and ELISA revealed that the same tumor-suppressing proteins, enriched in other MSC CM, were also elevated in CW-MSC CM (*Figure 4A–F*). Furthermore, MMP2 and MMP9, which promote the migration and invasion of OS cells, were downregulated in CW-MSC CM (*Figure 4A*). Using an MTT assay, the anti-tumor efficacy of the selected five tumor-suppressing proteins were evaluated with TT2 OS cells. The value $IC_{50}$ for CW-MSC CM was 390 μg/ml (*Figure 4G*), while $IC_{50}$ for the selected tumor-suppressing proteins ranged from 1.1 μg/ml (CALR) to 5.8 μg/ml (HSP) (*Figure 4H*).

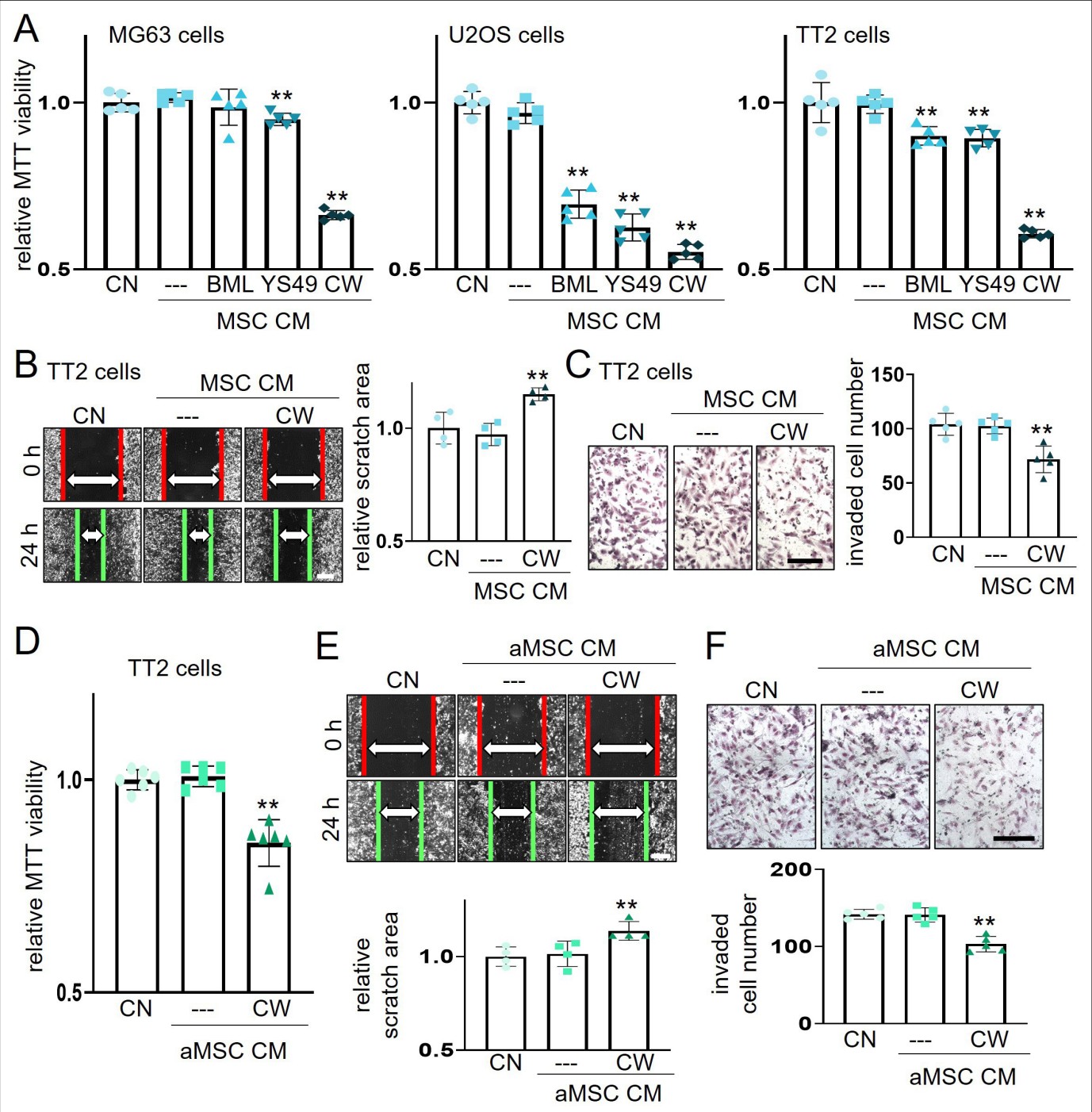

**Figure 1.** Suppression of the viability, migration, and invasion of osteosarcoma (OS) cells by CW008-treated mesenchymal stem cell (MSC) CM. The double asterisk indicates p<0.01. CN = control, CM = conditioned medium, BML = BML284 as a Wnt activator, YS49=PI3K activator, CW = CW008 as a PKA activator, MSC = bone marrow-derived MSC, and aMSC = adipose-derived MSC. (**A**) Reduction in MTT-based cell viability of three OS cell lines (MG63, U2OS, TT2 PDX) in 2 days by bone marrow-derived MSC CM, which were derived after the treatment with BML284, YS49, or CW008. (n=5). (B and C) Decrease in scratch-based motility (n=4) and transwell invasion (n=5) of TT2 OS cells in 2 days by CW008-treated MSC CM. (**D–F**) Inhibition of MT-based cell viability (n=6), scratch-based motility (n=4), and transwell invasion (n=5) of TT2 OS cells in 2 days by CW008-treated adipose-derived MSC. (Scale bar, 200 μm, error bars indicate standard deviation.)

The online version of this article includes the following figure supplement(s) for figure 1:

**Figure supplement 1.** Effects of H89 (PKA inhibitor) and D-cAMP (cAMP analog as a PKA activator).

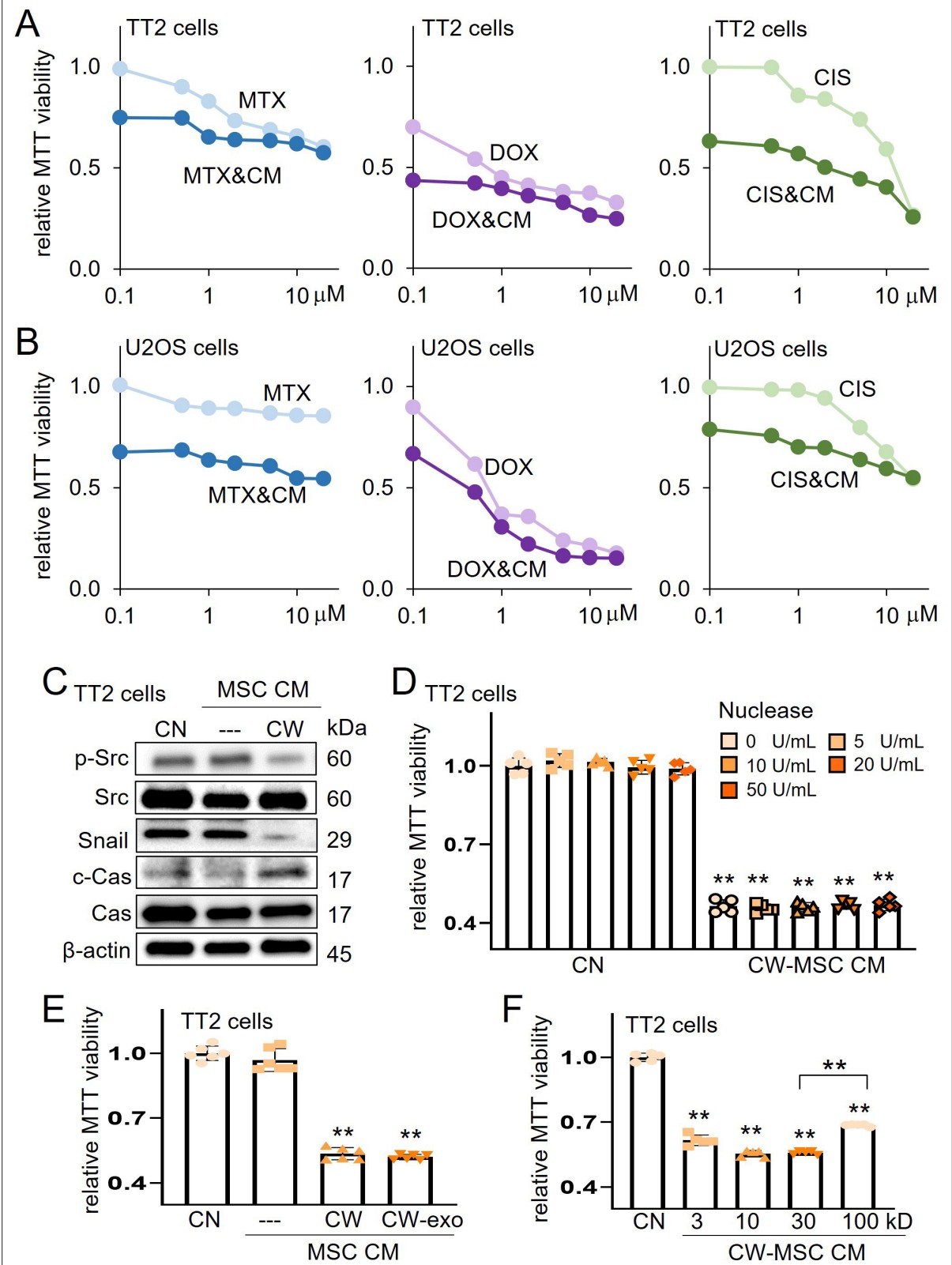

**Figure 2.** Characterization of CW008-treated mesenchymal stem cell (MSC) CM. The double asterisk indicated p<0.01. CN = control, CW = CW008, CM = conditioned medium, Cas = caspase 3, exo = exosome, MTX = methotrexate, DOX = doxorubicin, and CIS = cisplatin. (A and B) Additive MTT-based anti-tumor effect of CW008-treated MSC CM with methotrexate, doxorubicin, and cisplatin in TT2 and U2OS cells, respectively. (**C**) Reduction of p-Src and Snail and elevation of cleaved caspase 3 in TT2 OS cells by CW008-treated MSC CM for 2 days. (D and E) No significant change of MTT-based

*Figure 2 continued on next page*

*Figure 2 continued*

viability by the nuclease treatment (n=5) and ultracentrifugation for exosome removal (n=6), respectively, of CW008-treated MSC CM. (**F**) Variable tumor-suppressing capability of the size-fractionated CW008-treated bone marrow-derived MSC CM (CW-MSC CM) portion. (n=5). The protein size in kD on the X-axis indicates the cutoff size. For instance, the bar for 100 kD indicates the MTT value for the fraction that includes proteins larger than 100 kD. (Error bars indicate standard deviation.)

The online version of this article includes the following source data for figure 2:

**Source data 1.** Original files for the gels in *Figure 2C*.

## CALR's anti-tumor action on tumor-invaded bone

We have so far shown the tumor-suppressing capability of CW-MSC CM that contained the elevated level of the selected tumor-suppressing proteins. Since CALR was most significantly increased in CM (4.1×) and its IC$_{50}$ was the lowest among the five selected proteins, its OS-suppressing effect was tested in two groups of NSG mice, the placebo and CALR-treated groups. Mice (~8 weeks of age) received the inoculation of TT2 PDX cells in their proximal tibia by an intra-tibia injection, and CALR was administered at 10 µg/kg daily as a tail vein injection for 18 days. The placebo received a daily intravenous injection of the vehicle. X-ray images showed that compared to the placebo CALR-treated mice presented a smaller defect in the proximal tibia (*Figure 5A*), and the BMD of the cortical bone in the proximal tibia was higher (*Figure 5B*). Furthermore, microCT images of trabecular bone showed an increase in BV/TV and BMD in the proximal tibia (N=8, *Figure 5C*). Consistently, immunohisto-chemical analysis showed a decrease in Ki-67 and an increase in cleaved caspase 3 in tumor-invaded bone sections by the administration of CALR (*Figure 5—figure supplement 1*). Furthermore, the culturing of MC3T3 osteoblast cells in CW CM and CALR increased Alizarin Red staining in 3 weeks and elevated the levels of osteogenic genes such as type I collagen, alkaline phosphatase (ALP), and osteocalcin (*Figure 5—figure supplement 2*).

## Involvement of the CALR-CD47 regulatory axis

CALR is reported as a pro-phagocytic protein with CD47, a transmembrane integrin-associated protein as its ligand, which prevents cancer cell phagocytosis (*Chao et al., 2010*). Using a reciprocal pair of co-immunoprecipitations, we observed the interaction of CALR with CD47 using TT2 OS cell lysate (*Figure 5D*). We also observed using RNA interference that CALR-induced downregulation of p-Src and Snail, as well as upregulation of caspase 3, was suppressed by silencing *CD47* in U2OS cells (*Figure 5E*, *Figure 5—figure supplement 3*). Silencing *CD47* also downregulated MTT-based viability, whereas it significantly suppressed CALR-induced tumor inhibition in U2OS OS cells, TT2 OS cells, MDA-MB-231 breast cancer cells, and PANC1 pancreatic cells (*Figure 5—figure supplement 4*). In TCGA database, the overall survival of all cancer patients was favored with a high level of CALR and a low level of CD47 (*Figure 5F*).

## CALR's tumor-selective action

Expression of CD47 was elevated in three OS cell lines (U2OS, TT2, and MG63), compared to MSCs (*Figure 5G*). Consistently, CW-MSC CM and CALR preferentially inhibited the MTT-based viability of three OS cell lines. Of note, tumor selectivity was defined as a ratio of MTT values between cancer cells and non-cancer cells. Its values larger than 1 indicate the selective inhibition of OS cells (TT2, U2OS, and MG63) compared to non-OS cells (MSCs). By contrast, tumor selectivity for CIS, which was smaller than 1, indicated CIS's non-selective inhibition of tumor and non-tumor cells (*Figure 5H*). *CALR* is also known as a chaperone in the endoplasmic reticulum. We observed that the application of CALR recombinant protein to TT2 OS cells elevated the phosphorylation level of eukaryotic translation initiation factor 2 alpha (eIF2α), which regulates the stress to the endoplasmic reticulum (*Figure 5—figure supplement 5*).

## Double-sword role of the selected genes that were upregulated in OS

In the TCGA database, five transcripts for the selected tumor-suppressing proteins (*CALR, ENO1, HSP, MSN,* and *UBC*) were significantly upregulated in OS cells (*Figure 6A*). This observation raised an intriguing question as to whether highly expressed transcripts in OS cells could generate tumor-suppressing proteins. To test this counterintuitive hypothesis, we selected seven transcripts (*SPARC,*

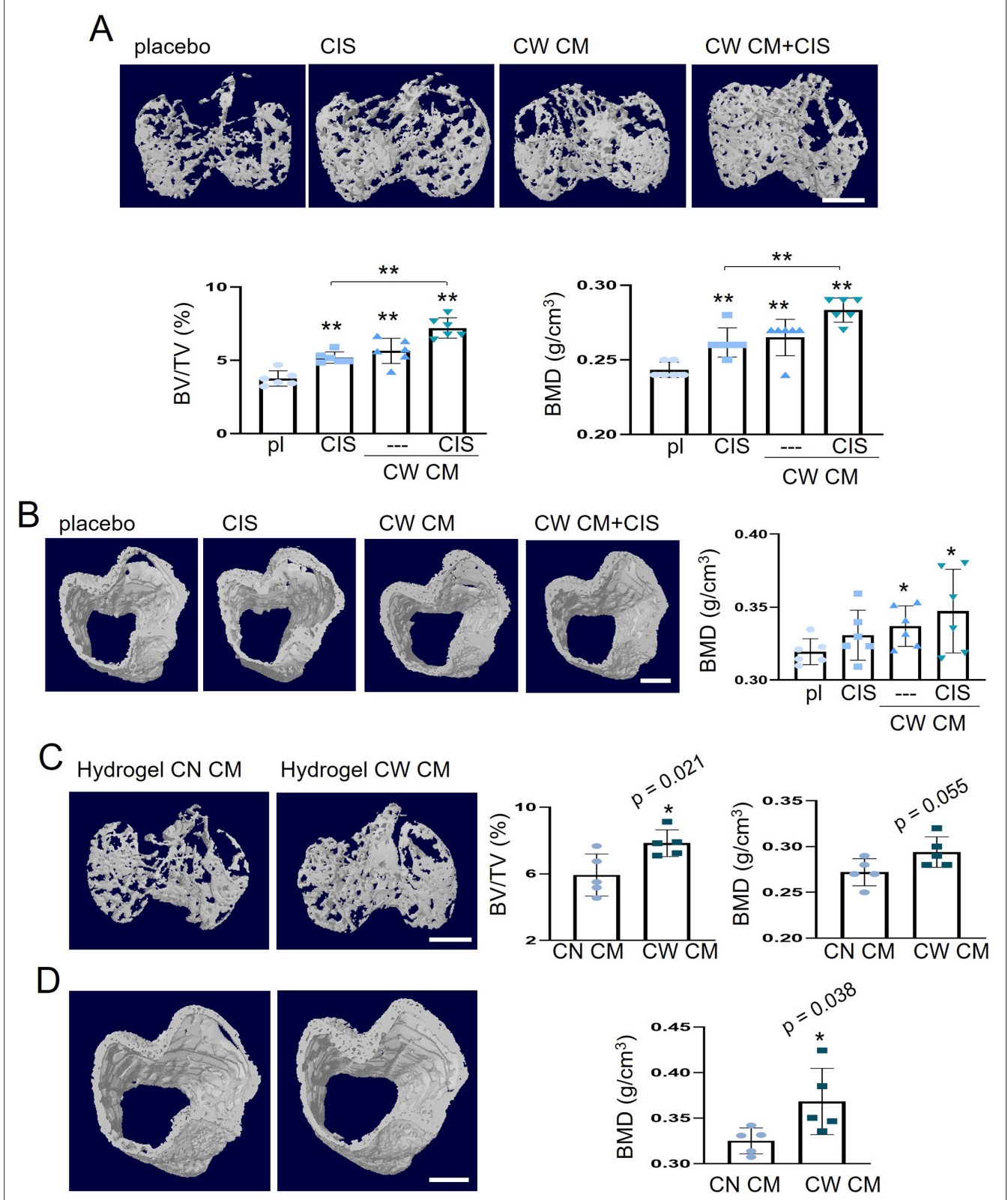

**Figure 3.** Protection of tumor-invaded bone by CW008-treated mesenchymal stem cell (MSC) CM. pl = placebo, CIS = cisplatin, CW = CW008, CN = control, and CM = conditioned medium. The single and double asterisks indicate p<0.05 and 0.01, respectively. (**A**) Additive prevention of bone loss in the tumor-invaded tibia by CW008-treated MSC CM with cisplatin (n=6). (**B**) Effect of CW008-treated MSC CM with cisplatin on tibial cortical bone

*Figure 3 continued on next page*

*Figure 3 continued*

(n=6). (C and D) Reduction in trabecular and cortical bone loss in the tumor-invaded proximal tibia by hydrogel CW008-treated MSC CM. BV/TV = bone volume ratio, BMD = bone mineral density (n=5). (Scale bar, 1 mm, error bars indicate standard deviation).

The online version of this article includes the following figure supplement(s) for figure 3:

**Figure supplement 1.** Histological analysis of tumor-invaded bone in response to CW008-treated MSC CM with and without cisplatin.

**Figure supplement 2.** Histological analysis of reduction in trabecular bone loss in the tumor-invaded proximal tibia by hydrogel CW008-treated mesenchymal stem cell (MSC) CM.

**Figure supplement 3.** Downregulation of Ki-67 and upregulation of cleaved caspase 3 in tumor-invaded bone sections by the administration of CW CM.

PCOLCE, CPE, GJA1, S100A11, HISTONE H4, and PPIB) that were highly expressed in OS cells (*Supplementary file 1*). An MTT assay revealed that all proteins except for SPARC (5 µg/ml) showed tumor-suppressing capability (*Figure 6B*). In CW-MSC CM, the levels of HISTONE H4, PPIB, and PCOLCE were elevated (*Figure 6C*). Because of the strongest effect of PCOLCE in the MTT assay, we focused on it and further analyzed its anti-tumor capability. The treatment with 1 µg/ml of PCOLCE reduced transwell invasion and scratch-based motility of TT2 OS cells (*Figure 6D and E*). Notably, a high level of PCOLCE significantly reduced the survival rate of all cancer patients (*Figure 6F*).

### Anti-tumor regulatory action of the PCOLCE-APP axis

According to the IntAct protein interaction database, PCOLCE was shown to interact with APP, a cell-surface protein. Interestingly, APP also interacts with CALR (*Figure 6G*). Notably, a high level of APP transcripts significantly reduced the survival rate of sarcoma patients (p=0.0051). However, its high level was not a negative factor for the survival of patients with all cancer (*Figure 6H*).

### Interaction of APP with PCOLCE and CALR

To assess the potential mechanism of tumor-suppressive action of PCOLCE and CALR, we conducted an immunoprecipitation assay. The result revealed that the APP was co-immunoprecipitated with PCOLCE and CALR in TT2 cell lysates (*Figure 7A*). APP acted oncogenic, and its silencing in TT2 cells reduced the transwell invasion and scratch-based motility (*Figure 7B–D*). Silencing APP also downregulated MTT-based viability, whereas it significantly suppressed PCOLCE/CALR-induced tumor inhibition (*Figure 7E*). Furthermore, the silencing of APP in TT2 cells suppressed PCOLCE-mediated downregulation of p-Src and Snail, and upregulation of cleaved caspase 3 (*Figure 7F*).

### Discussion

This study presented the generation of MSC-derived iTSCs by activating PKA signaling and evaluated the role of CALR-CD47 and PCOLCE-APP regulatory axes in suppressing the progression of OS (*Figure 7G*). In a traditional strategy, a gene with an elevated level of transcripts in cancer tissues is treated as a druggable target to be inhibited. In this study, we took a counterintuitive approach and showed the tumor-suppressive effects of their recombinant proteins. Together with the tumor-suppressing proteins such as CALR, ENO1, HSP, MSN, and UBC, which were enriched in tumor-suppressive CM, six recombinant proteins, such as HISTONE H4, PPIB, GJA1, CPE, S100A11, and PCOLCE, were identified as extracellular tumor-suppressing proteins. The results revealed a context-dependent role of proteins in CM and supported a paradoxical strategy to identify tumor-suppressing proteins. The study also demonstrated the hydrogel-based administration of CM to the site of tumor growth in the proximal tibia, and the compatibility of CM with CIS, a representative chemotherapeutic agent for OS treatment.

Among six tumor-suppressing proteins whose transcript levels were significantly elevated in OS tissues, two proteins (HISTONE H4 and PPIB) were enriched in iTSC CM in our previous studies (*Liu et al., 2021a*, *Sun et al., 2021*). HISTONE H4 is one of the five main histone proteins, and its extracellular form is cytotoxic via interactions with TLR2/4 (*Marsman et al., 2016*). PPIB catalyzes the cis-trans isomerization of peptide bonds, but little is known about its tumor-suppressive mechanism. Three proteins (CPE, S100A11, and PCOLCE) are reported tumorigenic in OS and/or other cancer. CPE is an enzyme to catalyze the release of C-terminal arginine or lysine residues, and its N-terminal truncated

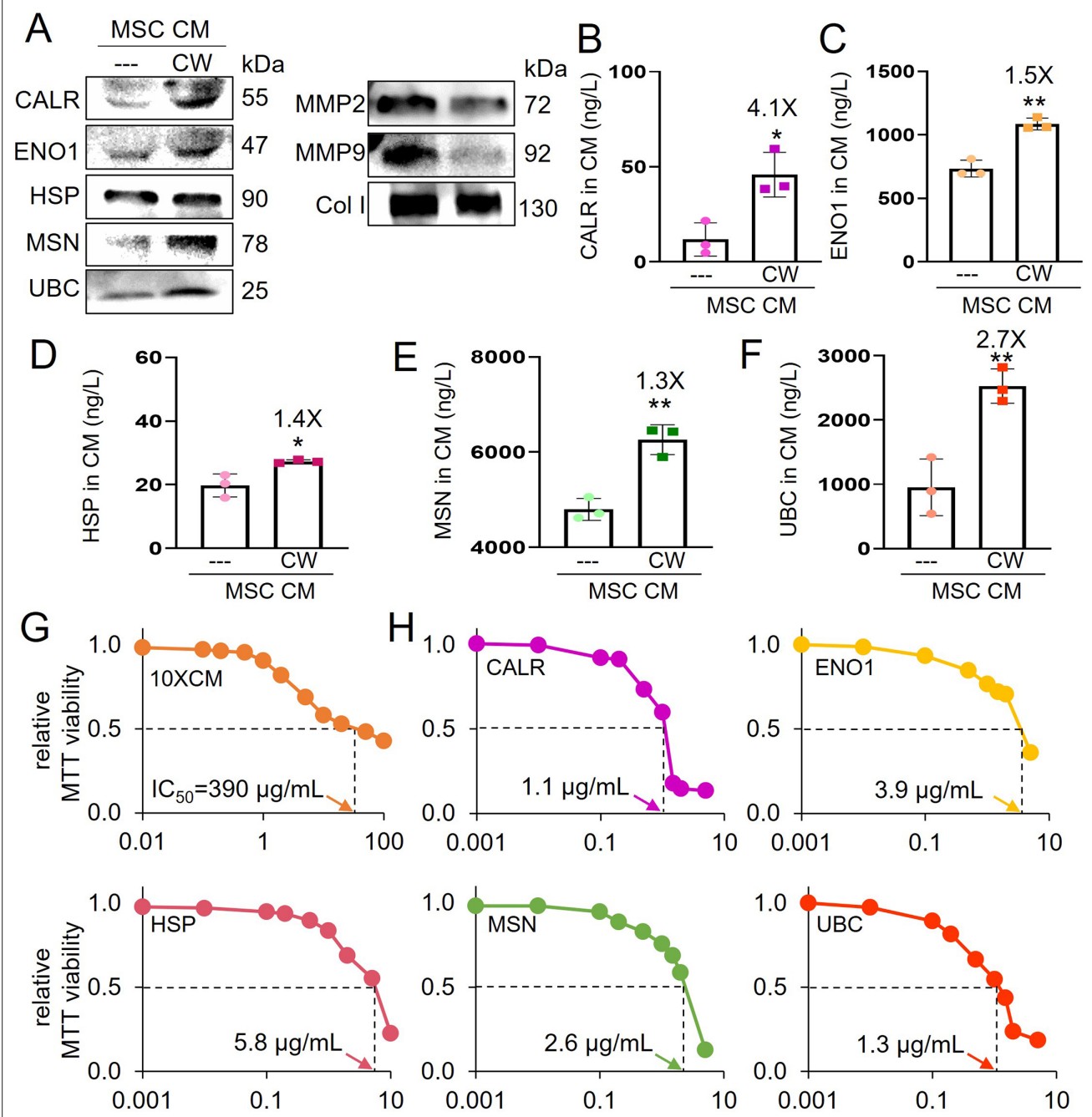

**Figure 4.** Tumor-suppressing proteins in CW008-treated mesenchymal stem cell (MSC) CM. The single and double asterisks indicate $p < 0.05$ and 0.01, respectively. CM = conditioned medium, CW = CW008 (PKA activator), CALR = calreticulin, ENO1=enolase 1, HSP = heat shock protein 90ab1, MSN = moesin, UBC = ubiquitin C, MMP2 and MMP9=matrix metalloproteinases 2 and 9, Col I=type I collagen. (**A**) Western blot-based expression levels of CALR, ENO1, HSP, MSN, UBC, MMP2, MMP9, and Col I in CW008-treated MSC CM. (**B–F**) ELISA-based levels of five tumor-suppressing proteins (CALR, ENO1, HSP, MSN, and UBC) in CW008-treated MSC CM (n=3). (**G**) Dose responses of TT2 OS cells in response to CW008-treated MSC CM with $IC_{50}$ at 390 µg/ml (n=5). (**H**) Dose responses and $IC_{50}$ of TT2 OS cells in response to five tumor-suppressing proteins (CALR, ENO1, HSP, MSN, and UBC). (Error bars indicate standard deviation.)

The online version of this article includes the following source data for figure 4:

**Source data 1.** Original files for the gels in *Figure 4A*.

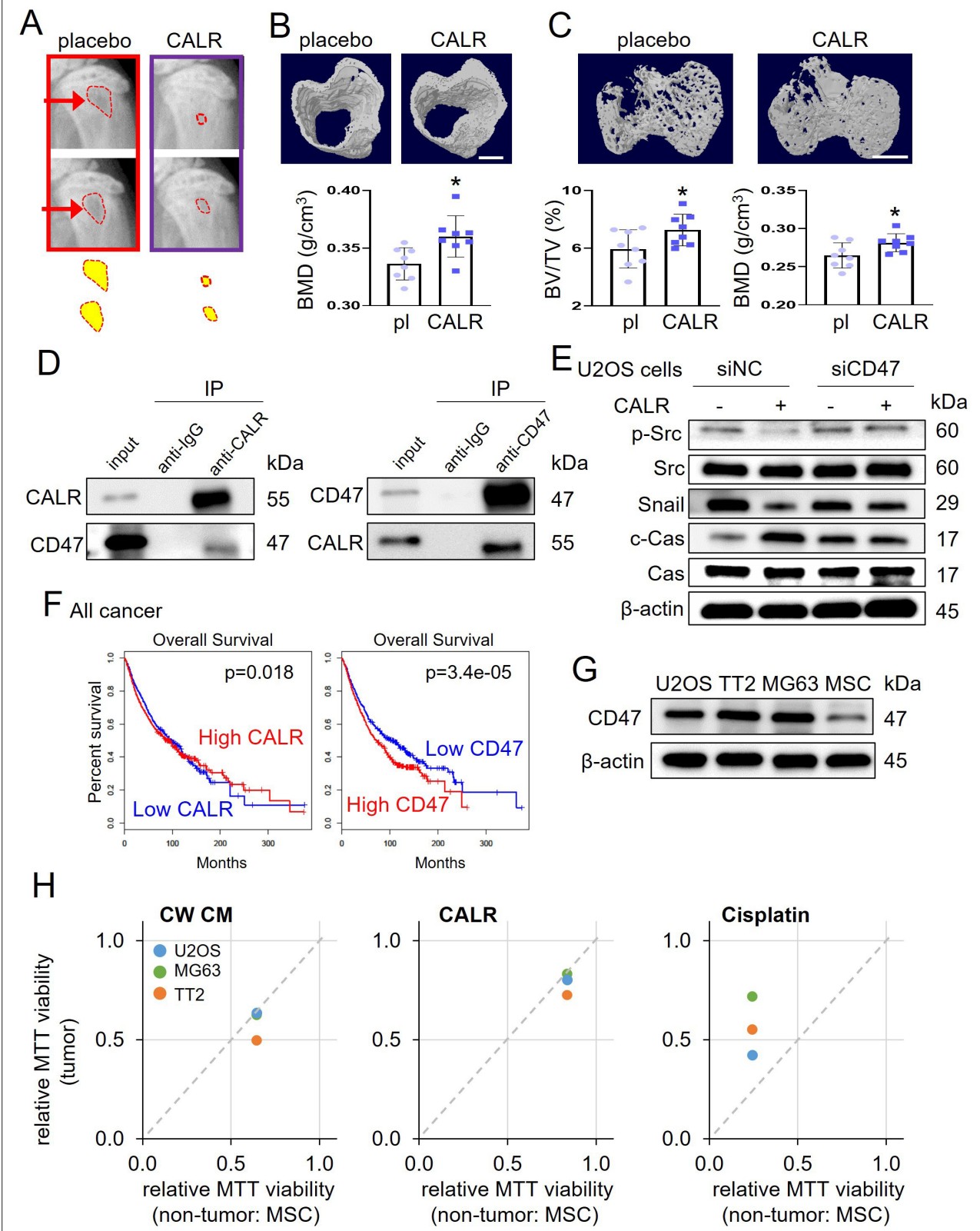

**Figure 5.** Calreticulin's action on tumor-invaded bone and its interaction with CD47. The single asterisk indicates p<0.05. CALR = calreticulin, pl = placebo, siNC = nonspecific siRNA, siCD47=CD47 siRNA, CW CM = CW008-treated MSC conditioned medium, and Cas = caspase 3. (**A**) X-ray images of the proximal tibia of NSG mice that received inoculation of TT2 OS cells. (**B**) MicroCT image-based increase in BMD (bone mineral density) of cortical bone in the proximal tibia (N=8) of NSG mice by daily injection of calreticulin (10 μg/kg). (**C**) MicroCT image-based increase in BV/TV (bone volume

*Figure 5 continued on next page*

*Figure 5 continued*

ratio) and BMD of trabecular bone of NSG mice in the proximal tibia (n=8). (**D**) Reciprocal co-immunoprecipitation of calreticulin and CD47 using TT2 osteosarcoma (OS) cell lysate. (**E**) Suppression of calreticulin-induced changes of p-Src, Snail, and cleaved caspase 3 (c-Cas) in U2OS cells by silencing CD47. (**F**) Favorable %survival with a high level of calreticulin and a low level of CD47 in all cancer patients. (**G**) Elevated expression level of CD47 in three OS cell lines (U2OS, TT2, and MG63), compared to non-OS cells (MSCs). (**H**) Tumor selectivity, larger than 1 for CW008-treated MSC CM and calreticulin, indicating the selective inhibition of OS cells (TT2, U2OS, and MG63) compared to MSCs. Of note, tumor selectivity for cisplatin, smaller than 1, indicates cisplatin's non-selective inhibition of tumor and non-tumor cells. (Scale bar, 1 mm, error bars indicate standard deviation.)

The online version of this article includes the following source data and figure supplement(s) for figure 5:

**Source data 1.** Original files for the gels in *Figure 5D, E and G*.

**Figure supplement 1.** Downregulation of Ki-67 and upregulation of cleaved caspase 3 in tumor-invaded bone sections by the administration of calreticulin.

**Figure supplement 2.** Stimulation of osteoblast differentiation by CW008-treated bone marrow-derived MSC CM (CW-MSC CM) and calreticulin.

**Figure supplement 2—source data 1.** Original files for the gels in *Figure 5—figure supplement 2B*.

**Figure supplement 3.** Silencing the CD47 in U2OS cells.

**Figure supplement 3—source data 1.** Original files for the gels in *Figure 5—figure supplement 3*.

**Figure supplement 4.** Suppression of CALR-driven decrease in MTT-based viability of U2OS osteosarcoma (OS) cells, TT2 OS cells, MDA-MB-231 breast cancer cells, and PANC1 pancreatic cells by partial silencing of CD47.

**Figure supplement 5.** Elevation of the phosphorylation of eukaryotic translation initiation factor 2 alpha (p-eIF2a) in TT2 osteosarcoma (OS) cells by the application of calreticulin (CALR).

**Figure supplement 5—source data 1.** Original files for the gels in *Figure 5—figure supplement 5*.

form promotes the migration and invasion of OS cells via Wnt signaling (*Fan et al., 2019*). S100A11 is reported to interact with miR-22 in MG63 OS cells, and it suppresses the anti-tumor action of miR-22 (*Zhou et al., 2018*). PCOLCE binds to type I procollagen, and its upregulation is reported to promote metastasis in OS (*Wang et al., 2019*). The role of GJA1, a member of the connexin family, is not well understood in OS progression. Collectively, the result clearly showed that tumorigenic transcripts, highly expressed in OS, frequently generate tumor-suppressing proteins in the extracellular domain.

The result revealed the significant role of the PCOLCE-APP regulatory axis in blocking OS progression. As a procollagen processing enzyme, PCOLCE is involved in the development of collagenous tissues, and its upregulation may induce an increase in collagen deposition and pathogenic fibrosis (*Kessler and Hassoun, 2019*). It thus plays a role in the transformation of the tumor microenvironment. Its marked elevation in OS tissues is reported to promote lung metastasis (*Wang et al., 2019*), and it is also upregulated in gastric cancer (*Xiang et al., 2020*). APP is an integral membrane protein, expressed mostly in neuronal tissues. It is a precursor of amyloid beta that contributes to producing amyloid plaques found in the brains of Alzheimer's disease patients. APP is also reported as one of the diagnostic biomarkers for OS (*Zhang and Yang, 2018*), and its high expression reduces the survival of OS patients but not pan-cancer patients. Taken together, APP can be a druggable target, selective to OS, and PCOLCE is one of its biological blockers.

In the CALR-CD47 regulatory axis, the results in this study revealed that extracellular CALR inhibited CD47's tumorigenic actions. CALR is a $Ca^{2+}$-binding endoplasmic reticulum protein and is also detected on the cell surface as well as in extracellular space (*Arosa et al., 1999*), and CD47 is an integrin-associated transmembrane protein. Although the role of CALR is reported context-dependent (*Fucikova et al., 2021*), it in general oppositely mediates phagocytosis. A pro-phagocytic signal of CALR is counteracted by CD47 (*Chao et al., 2010*), and blockage of CD47 is reported to induce in vivo tumor elimination by stimulating phagocytosis of cancer cells (*Chao et al., 2010*). Consistently, the inhibition of CD47 is reported to block the progression of OS in a mouse model (*Xu et al., 2015*). In this study, the level of CALR was upregulated 4.1-fold in CW-MSC CM, and the level of CD47 was higher in three OS cell lines than MSCs and osteocytes.

Translationally, the application of CM for any cancer treatment is far from mainstream compared to chemotherapeutic drugs. Its potential advantage is its integrity as a therapeutic agent with thousands of protein species. Alternatively, the selection of potent tumor-suppressing proteins such as CALR, ENO1, PCOLCE, etc. can be another option to construct a protein cocktail. The cocktail proteins can be developed based on the transcriptome of OS tissues of individual patients. The advantage of the use of MSCs is their immune privilege, and allogenic MSCs may not elicit inflammatory responses,

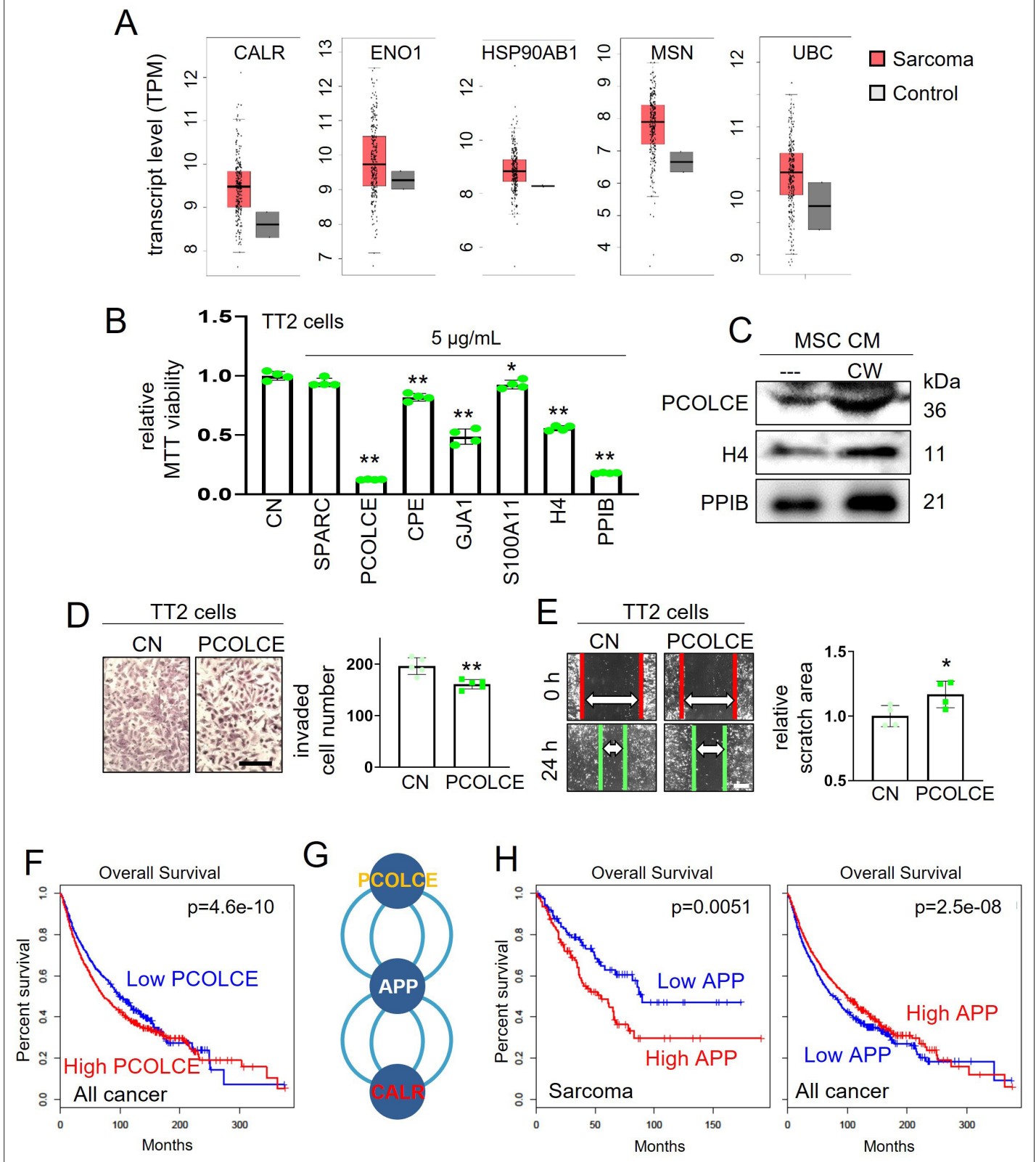

**Figure 6.** TCGA database-based prediction of tumor suppressors and the effect of PCOLCE. CN = control and APP = amyloid precursor protein. The single and double asterisks indicate p<0.05 and 0.01, respectively. (**A**) List of five selected proteins that were expressed higher in sarcoma tissues than the normal tissues in the TCGA database. (**B**) Reduction in MTT-based proliferation of TT2 osteosarcoma (OS) cells by seven selected recombinant proteins (5 µg/ml) in 48 hr, which are selected with higher expression in sarcoma tissues in the TCGA database. (**C**) Western blot images, showing that

*Figure 6 continued on next page*

*Figure 6 continued*

the levels of H4, PPIB, and PCOLCE were elevated in CW008-treated bone marrow-derived MSC CM (CW-MSC CM). (D and E) Inhibition in the transwell invasion (48 hr) and scratch-based migration (24 hr) of TT2 OS cells by 1 µg/ml PCOLCE recombinant protein. (**F**) Lowered survival with a high level of PCOLCE in all cancer patients. (**G**) Protein candidates, which are connected with PCOLCE by IntAct network analysis. (**H**) Lowered survival rates of sarcoma patients with high levels of APP, while the opposite effect of all cancer types. (Scale bar, 200 µm, error bars indicate standard deviation.)

The online version of this article includes the following source data for figure 6:

**Source data 1.** Original files for the gels in *Figure 6C*.

mainly due to their lack of class-II major histocompatibility complex (*Ankrum et al., 2014*). We mainly used bone marrow-derived MSCs in this study, but adipose tissue-derived MSCs were also able to generate iTSCs and tumor-suppressive CM.

In this study, we presented the marked anti-OS action of PKA-activated MSC CM and CALR, as well as CM's compatibility with CIS in a mouse model of OS. Although MSC CM contains other biomolecules such as nucleic acids, small metabolites, lipids, etc., we focused herein on proteomes since the treatment with nucleases, a removal of exosomes by ultracentrifugation, and filtering with 3 kD cutoff did not significantly alter CM's anti-tumor action. Also, we mainly employed CW008 and activated PKA signaling, but CW008 is known to promote osteogenic differentiation of MSCs (*Kim et al., 2013*) which may indirectly affect the composition of MSC CM. Lastly, the most frequent site of OS-associated metastasis is the lung. We have previously shown that the administration of PI3K-activated MSC-derived CM suppressed breast cancer-associated lung metastasis using a mouse model (*Sun et al., 2022a*). Besides the inhibition of the primary bone tumor, it is recommended to evaluate CM's inhibitory action for the metastasized tumor in the lung.

The study disclosed an uncommon use of OS-elevated transcripts as a candidate for producing OS-inhibiting proteins. In addition to tumor-suppressing proteins that were enriched in CW-MSC CM such as CALR, we identified PCOLCE as a novel tumor-suppressing protein and proposed a regulatory axis of PCOLCE-APP for the treatment of OS. In summary, we demonstrated that bone marrow-derived MSCs can be converted into iTSCs and their proteomes can be used to protect the bone, in a combinatorial application with CIS. The therapeutic CM was also given locally to the tibia using a PADO hydrogel-based delivery system. The PADO hydrogels were injectable, cured in situ, and degraded within 1 week in vivo (*Lin et al., 2022*). We envision that the described approach can be used to identify cell-surface receptors as druggable targets that mediate CM's anti-tumor action.

## Materials and methods

### Cell culture and agents

MG63 and U2OS human OS cell lines (#86051601-1VL and #92022711-1VL, Sigma, St Louis, MO, USA), and a patient-derived xenograft (PDX) TT2-77 xenoline were grown in DMEM (*Pandya et al., 2020*). Human MSCs (#PT-2501, Lonza, Basel, Switzerland) and human aMSCs (#SCC038, Sigma) were grown in MSCBM (#PT-3238, Lonza). MDA-MB-231 breast cancer cells (ATCC, Manassas, VA, USA) were grown in αMEM. MC3T3 osteoblasts (Sigma, St Louis, MO, USA), and PANC1 (ATCC) were grown in DMEM. The culture media were supplemented with 10% fetal bovine serum (FBS) and antibiotics (100 units/ml penicillin, and 100 µg/ml streptomycin; #15140122, Life Technologies, Grand Island, NY, USA), and cells were maintained at 37°C and 5% $CO_2$. The identity of MG63, U2OS, MDA-MB-231, MC3T3, and PANC1 cell lines was authenticated by DNA fingerprinting using a cell line authentication kit (ATCC), and TT2-77 xenoline was authenticated by IDEXX BioAnalytics. All these cell lines tested negative for mycoplasma contamination.

The recombinant proteins, including CALR (MBS2009125), ENO1 (MBS2009113), MSN (MBS2031729), UBC (MBS2029484, MyBioSource, San Diego, CA, USA), and HSP(OPCA05157, Aviva Systems Biology, San Diego, CA, USA) were given to TT2 cells, and cells were incubated for 48 hr. The activators for Wnt signaling (BML284, 0.2 µM, SC-222416, Santa Cruz, Dallas, TX, USA) (*Wang et al., 2009*), PI3K/Akt (YS-49, 50 µM, HY-15477, MCE, NJ, USA) (*Hsu et al., 2014*), and PKA signaling (CW008, 20 µM, #5495, Tocris, Minneapolis, MN, USA) (*Kim et al., 2013*) were applied to cells for 24 hr.

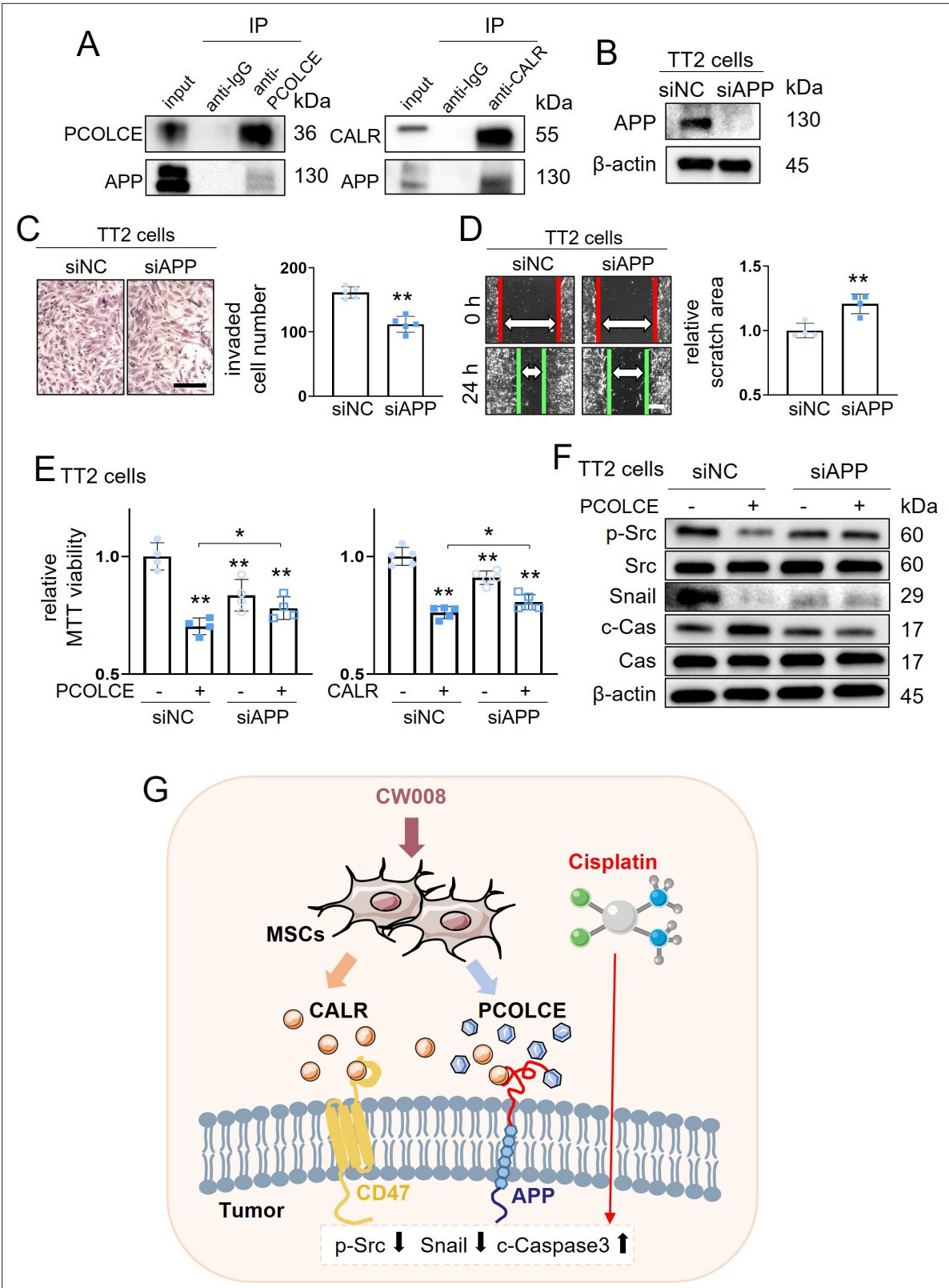

**Figure 7.** Putative regulatory mechanism for the tumor-suppressing capability of CW008-treated mesenchymal stem cell (MSC) conditioned medium (CM). CALR = calreticulin, siNC = nonspecific siRNA, siAPP = APP siRNA, and Cas = caspase 3. The single and double asterisks indicate p<0.05 and 0.01, respectively. (**A**) Co-immunoprecipitation of amyloid precursor protein (APP) with PCOLCE and CALR. (**B–D**) Inhibition in the transwell invasion (48 hr) (n=5) and scratch-based migration (24 hr) (n=4) of TT2 osteosarcoma (OS) cells by silencing the APP. (**E**) Suppression of PCOLCE (n=4) and CALR

*Figure 7 continued on next page*

*Figure 7 continued*

(n=5)-driven decrease in MTT-based viability of TT2 OS cells by partial deletion of APP. (**F**) Suppression of PCOLCE-induced changes of p-Src and cleaved caspase 3 (c-Cas) in TT2 cells by silencing APP. (**G**) Schematic diagram for the regulatory mechanism of tumor-suppressing action of CW008-treated MSC CM. (Scale bar, 200 µm, error bars indicate standard deviation.)

The online version of this article includes the following source data for figure 7:

**Source data 1.** Original files for the gels in *Figure 7A, B and F*.

## Preparation of CM

For in vitro experiments, CM was subjected to low-speed centrifugation at 2000 rpm for 10 min. The cell-free supernatants were centrifuged at 4000 rpm for 10 min and subjected to filtration with a 0.22 µm polyethersulfone membrane (#SLGPR33RS, Sigma, St Louis, MO, USA). The supernatants were further centrifuged at $10,000 \times g$ for 30 min at 4°C to remove remaining cell debris and at $100,000 \times g$ (Type 90 Ti Rotor, Beckman, Brea, CA, USA) overnight at 4°C to remove exosomes. For examining the efficacy of CM in vivo, FBS-free CM was condensed by a filter with a cutoff molecular weight of 3 kD, and the condensed CM (50 µl resuspended in PBS) was intravenously injected from the tail vein. To evaluate the effect of nucleic acids on the anti-tumor action of CM, we treated CM with nucleases for digesting DNA and RNA (#PI88700, Thermo Fisher Scientific, Waltham, MA, USA). We also used four filters with different cutoff weights of 3, 10, 30, and 100 kD (#UFC900324, #UFC8010, #UFC8030, #UFC8100, Sigma) and evaluated the anti-tumor efficacy of the size-fractionated CMs.

## MTT assay

MTT-based metabolic activity was evaluated using ~2000 cells seeded in 96-well plates (#CLS3585, Corning, Glendale, AZ, USA). CM or recombinant proteins was given on day 2, and cells were dyed with 0.5 mg/ml thiazolyl blue tetrazolium bromide (#M5655-1G, Sigma) on day 4 for 4 hr. Optical density for assessing metabolic activities was determined at 562 nm using a multi-well spectrophotometer.

## Transwell invasion assay

The invasion capacity of OS cells was determined using a 12-well plate and transwell chambers (#353182, Thermo Fisher Scientific) with 8 µm pore size. Transwell chambers were coated with 300 µl Matrigel (#354234, Thermo Fisher Scientific, 100 µg/ml) that was polymerized and dried overnight. Five hundred µl of the serum-free medium was added to each chamber and after 1 hr, the chamber was washed three times with the serum-free medium. Approximately $7 \times 10^4$ cells in 300 µl serum-free DMEM were then placed in the upper chamber and 800 µl CM was added to the lower chamber. After 48 hr, the cells on the upper surface of the membrane were removed. The cells, which invaded the lower side of the membrane, were fixed and stained with 100% methanol and Crystal Violet (diluted 1:25 in water) for 30 min. At least five randomly chosen images were taken with an inverted optical microscope (magnification, ×100, Nikon, Tokyo, Japan), and the average number of stained cells, which represented the invasion capacity, was determined.

## Two-dimensional motility scratch assay

A motility scratch assay was conducted to evaluate two-dimensional migratory behavior. Approximately $4 \times 10^5$ cells were seeded in 12-well plates. After cell attachment, a plastic pipette tip was used to scratch a gap on the cell layer. The floating cells were removed, and CM or recombinant protein was added. Images of the cell-free scratch zone were taken at 0 hr, and the areas newly occupied with cells were determined 24 hr after the scratching. The areas were quantified with Image J (National Institutes of Health, Bethesda, MD, USA).

## Western blot analysis and ELISA

Cultured cells were lysed in a radio-immunoprecipitation assay buffer. Proteins were fractionated by 10–15% SDS gels and electro-transferred to polyvinylidene difluoride transfer membranes (#IPVH00010, Millipore, Billerica, MA, USA). The membrane was incubated overnight with primary antibodies and then with secondary antibodies conjugated with horseradish peroxidase (#7074S and #7076S, Cell Signaling, Danvers, MA, USA). Antibodies against p-Src (#2105S), Src (#2108S), p-eIF2α (#9721S), eIF2α (#9722S), Snail (#3879S), cleaved caspase 3 (#9661S), caspase 3 (#9662S), CALR

(#2891S), ENO1 (#3810S), MSN (#3146S), APP (#2452S), MMP2 (#87809S) (Cell Signaling), MMP9 (#sc-393859), PCOLCE (#sc-73002), ALP (#sc-28904) (Santa Cruz, Dallas, TX, USA), HSP, osteocalcin (#ab203085, #ab93876, Abcam, Boston, MA, USA), CD47 (#MA5-11895), UBC (#PA5-76144) (Thermo Fisher Scientific), and type I collagen (#NB600-408, Novus Biologicals, CO, USA) were employed with β-actin as a control (#A5441, Sigma). The level of proteins was determined using a SuperSignal west femto maximum sensitivity substrate (#34096, Thermo Fisher Scientific), and a luminescent image analyzer (LAS-3000, Fuji Film, Tokyo, Japan) was used to quantify signal intensities. The levels of CALR, ENO1, HSP, MSN, and UBC in CW008-treated CM were determined using the ELISA kits (MBS263181, MBS1604563, MBS7700502, MBS2709503, and MBS267048; MyBioSource).

## RNA interference
RNA interference was conducted using siRNA specific to *CD47* and *APP* (#s1501, #145977; Thermo Fisher Scientific) with a negative siRNA (Silencer Select #1, Thermo Fisher Scientific) as a nonspecific control using the procedure previously described (*Minami et al., 2017*).

## Immunoprecipitation
Immunoprecipitation was conducted with an immunoprecipitation starter pack kit (#45-000-369, Cytiva, Marlborough, MA, USA), using the procedure the manufacturer provided. In brief, 20 µl of protein A sepharose was washed twice with PBS and incubated with 2 µg of antibodies for CALR, CD47, or PCOLCE. In parallel, control IgG was prepared for negative control. The antibody cross-linked beads were incubated overnight with 600 µl TT2 cell protein samples on a shaker. The beads were collected by centrifugation, washed three times with PBS, and resuspended for western blotting. The protein samples before the immunoprecipitation were used as positive controls. Western blotting was conducted using antibodies against CALR, CD47, PCOLCE, and APP.

## Differentiation of osteoblasts
To evaluate the effect of CW-MSC CM and CALR recombinant protein on the differentiation of osteoblasts, MC3T3 osteoblasts were cultured in the osteogenic medium that consisted of 50 µg/ml ascorbic acid and 10 mM sodium β-glycerophosphate with 10% FBS and antibiotics. The medium was exchanged every 3 days, and cells were fixed and stained with Alizarin Red to visualize calcium deposits in 3 weeks.

## Immunohistochemistry
Using immunohistochemical staining, we detected the expression levels of Ki-67 (#ab16667, Abcam) and cleaved caspase 3 in the sagittal sections of the proximal tibia approximately 2 mm distal to the growth plate. After deparaffinization, the sections were placed in a 10 mM citric acid buffer for antigen activation. The sections were incubated with primary antibodies, followed by an ALP-conjugated secondary antibody, which was used for chromogenic staining. The sections were counterstained with hematoxylin. The immunostained area was quantified in the tumor-invaded area.

## Prediction of tumor-suppressing proteins from TCGA database
Seven tumor-suppressing protein candidates, including SPARC, histone H4, PPIB, GJA1, CPE, S100A11, and PCOLCE, were predicted from the transcriptomes in the TGCA database. The transcript levels of these candidates were elevated at least two times in OS tissues. We previously showed that extracellular histone H4 and PPIB acted as tumor-suppressing proteins in breast cancer (*Liu et al., 2021a*, *Sun et al., 2021*), while the five others were reported to stimulate the progression of OS when present in the intracellular domain (*Chen et al., 2021*; *Wang et al., 2019*; *Zhang et al., 2021*; *Talbot et al., 2020*; *Zhou et al., 2018*). To examine anti-tumor capabilities, we employed the recombinant proteins (MBS553651, MBS2097677, MBS2009092, MBS2029317, MBS146520, MBS2557159, MBS2031097; MyBioSource), respectively. In the MTT assay, 5 µg/ml of each of these recombinant proteins was added to the medium and the alteration in the metabolic activity of OS cells was evaluated in 2 days.

## Animal models
The experimental procedures were approved by the Indiana University Animal Care and Use Committee and complied with the Guiding Principles in the Care and Use of Animals endorsed by the American

Physiological Society. Mice were randomly housed five per cage by a stratified randomization procedure based on body weight. Mouse chow and water were provided ad libitum. In the mouse model of osteolysis, NOD/SCID/γ (-/-) (NSG) female mice (~8 weeks) were provided by the In Vivo Therapeutics Core of the Indiana University Simon Comprehensive Cancer Center (Indianapolis, IN, USA). Mice were divided into four groups (placebo, CIS, CW CM, and CW CM and CIS groups; six mice per group) in the first experiment, while they were divided into two groups (hydrogel CN CM and hydrogel CW CM groups; five mice per group) in the second experiment. They received an injection of TT2 cells ($2.5 \times 10^5$ cells in 20 µl PBS), into the right tibia as an intra-tibial injection. From day 2, mice in the CM treatment groups received a daily intravenous injection of 50 µl CW-treated MSC CM, while CIS treatment groups received a weekly intraperitoneal injection of 10 µg/kg. The hydrogel groups received a weekly subcutaneous injection of 50 µl PADO hydrogel with lyophilized CM powder. For examining the efficacy of CALR, NSG female mice were divided into two groups (placebo and CALR groups; eight mice per group) and received an intra-tibial injection of TT2 cells ($2.5 \times 10^5$ cells in 20 µl PBS) to the right tibia on day 1. From day 2, mice were given a daily intravenous injection of PBS (placebo group) or 10 µg/kg of CALR recombinant protein (CALR group). The animals were sacrificed on day 18, and the hindlimbs were harvested for microCT imaging and histology.

## Injectable hydrogel fabrication

Poly (diacetone acrylamide-co-di (ethylene glycol) ethyl ether acrylate-co-oligo (ethylene glycol) methyl ether acrylate), or PADO, was synthesized using reversible addition-fragmentation chain-transfer (RAFT) polymerization as previously reported (*Lin et al., 2022*). The monomers feed ratios were 29, 59, and 13 mol% for diacetone acrylamide, di (ethylene glycol) ethyl ether acrylate, and oligo (ethylene glycol) methyl ether acrylate, respectively. PADO was crosslinked via acyl hydrazone chemistry using adipic dihydrazide (ADH) as the crosslinker. In addition to the acyl hydrazone crosslink, PADO was physically crosslinked due to phase separation occurring above room temperature. For conditioned media encapsulation, 15 wt% PADO was first reacted with ADH with hydrazide to ketone ratio of one overnight. Lyophilized conditioned media powder was reconstituted using low glucose DMEM and added to the polymer solution. Prior to injection, PADO hydrogel precursor solution was sterilized using germicidal UV light and then 50 µl was injected into each leg.

## X-ray

Whole-body X-ray imaging was performed using a Faxitron radiographic system (Faxitron X-ray Co., Tucson, AZ, USA) (*Bonar et al., 2012*). Tibial integrity was scored in a blinded manner on a scale of 0–3: 0 = normal with no indication of a tumor, 1 = clear bone boundary with slight periosteal proliferation, 2 = bone damage and moderate periosteal proliferation, and 3 = severe bone erosion.

## µCT imaging and histology

The tibiae were harvested for µCT imaging and histology. Micro-computed tomography was performed using Skyscan 1172 (Bruker-MicroCT, Kontich, Belgium) (*Bouxsein et al., 2010*). Using manufacturer-provided software, scans were performed at pixel size 8.99 µm and the images were reconstructed (nRecon v1.6.9.18) and analyzed (CTan v1.13). Using µCT images, trabecular bone parameters such as BV/TV and BMD were determined in a blinded fashion, as well as cortical bone parameters such as BMD. In histology, H&E staining was conducted as described previously and images were analyzed in a blinded fashion (*Fan et al., 2020*). Of note, normal bone cells appeared in a regular shape with round and deeply stained nuclei, while tumor cells were in a distorted shape with irregularly stained nuclei.

## Statistical analysis

For cell-based experiments, three or four independent experiments were conducted, and data were expressed as mean ± SD. In animal experiments, we employed at least five mice per group to obtain statistically significant differences in BV/TV and BMD for trabecular bone and BMD for cortical bone as a primary outcome measure. Statistical significance was evaluated using a one-way analysis of variance (ANOVA). Post hoc statistical comparisons with control groups were performed using Bonferroni correction with statistical significance at $p < 0.05$. The single and double asterisks in the figures indicate $p < 0.05$ and $p < 0.01$, respectively.

## Acknowledgements

The authors appreciated the Indiana Pediatrics Biobank for the collection and processing of TT2 OS tissues and cells.

## Additional information

### Funding

| Funder | Grant reference number | Author |
| --- | --- | --- |
| Biomechanics and Biomaterials Research Center at Indiana University Purdue University Indianapolis, USA | NO. 2201-01 | Hiroki Yokota |
| Eunice Kennedy Shriver National Institute of Child Health & Human Development | NO. P50HD090215 | Karen E Pollok |
| National Cancer Institute | NCI Cancer Center Support Grant NO. P30CA082709 | Karen E Pollok |
| Tyler Trent Cancer Research Endowment for the Riley Hospital for Children IU-Health, USA | | Karen E Pollok |
| Indiana University | Grand Challenge-Precision Health Initiative | Karen E Pollok |

The funders had no role in study design, data collection and interpretation, or the decision to submit the work for publication.

### Author contributions

Kexin Li, Conceptualization, Data curation, Formal analysis, Investigation, Writing – original draft, Writing – review and editing; Qingji Huo, Nathan H Dimmitt, Guofan Qu, Junjie Bao, Data curation, Formal analysis, Investigation, Writing – review and editing; Pankita H Pandya, Formal analysis, Investigation, Writing – review and editing; M Reza Saadatzadeh, Khadijeh Bijangi-Vishehsaraei, Resources, Formal analysis, Investigation, Writing – review and editing; Melissa A Kacena, Conceptualization, Investigation, Writing – original draft, Writing – review and editing; Karen E Pollok, Resources, Supervision, Funding acquisition, Investigation, Writing – review and editing; Chien-Chi Lin, Supervision, Funding acquisition, Investigation, Writing – review and editing; Bai-Yan Li, Conceptualization, Supervision, Investigation, Writing – original draft, Project administration, Writing – review and editing; Hiroki Yokota, Conceptualization, Supervision, Funding acquisition, Investigation, Writing – original draft, Project administration, Writing – review and editing

### Author ORCIDs

Qingji Huo  http://orcid.org/0000-0003-0438-6812
Khadijeh Bijangi-Vishehsaraei  http://orcid.org/0009-0009-2522-8806
Melissa A Kacena  http://orcid.org/0000-0001-7293-0088
Chien-Chi Lin  http://orcid.org/0000-0002-4175-8796
Hiroki Yokota  http://orcid.org/0000-0002-7881-8959

### Ethics

All animal experiments were conducted according to protocols approved by the Indiana University Animal Care and Use Committee and complied with the Guiding Principles in the Care and Use of Animals endorsed by the American Physiological Society (protocol #345R).

## Decision letter and Author response

Decision letter https://doi.org/10.7554/eLife.83768.sa1
Author response https://doi.org/10.7554/eLife.83768.sa2

# Additional files

## Supplementary files
• Supplementary file 1. List of seven selected proteins that were expressed higher in sarcoma tissues than the normal tissues in the TCGA dataset.

• MDAR checklist

## Data availability
Data available at Dryad: https://doi.org/10.5061/dryad.m905qfv4w.

The following dataset was generated:

| Author(s) | Year | Dataset title | Dataset URL | Database and Identifier |
|---|---|---|---|---|
| Li K, Huo Q, Dimmitt NH, Qu G, Bao J, Pandya PH, Saadatzadeh MR, Bijangi-Vishehsaraei K, Kacena MA, Pollok KE, Lin C-C, B-Y Li, Yokota H | 2023 | Osteosarcoma-enriched transcripts paradoxically generate osteosarcoma-suppressing extracellular proteins | https://doi.org/10.5061/dryad.m905qfv4w | Dryad Digital Repository, 10.5061/dryad.m905qfv4w |

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
