## [Editor Report]

There are no known effective treatments available to date for the treatment of osteosarcomas, the earliest identified bone cancer that can spread to other tissues. In this study, the authors have used novel approaches to identify calreticulin and procollagen C-endopeptidase enhancer (PCOLCE) as osteosarcoma tumor suppressor proteins that inhibit osteosarcoma growth both in animal and in vitro cell culture models. These important findings may provide a basis for the future development of more efficient targeted therapies for the treatment of osteosarcomas.

---

## [Decision Letter]

**Decision letter after peer review:**

Thank you for submitting your article "Osteosarcoma-enriched transcripts paradoxically generate osteosarcoma-suppressing extracellular proteins" for consideration by *eLife*. Your article has been reviewed by 2 peer reviewers, and the evaluation has been overseen by a Reviewing Editor and Mone Zaidi as the Senior Editor. The following individual involved in the review of your submission has agreed to reveal their identity: Haibo Zhao (Reviewer #1).

Essential revisions:

1) The increases in both trabecular and cortical bone mass in PKA-induced MSC conditional medium (Figure 3) and recombinant calreticulin (Figure 5A-5C) treated mice may be in part due to enhanced osteoblast differentiation from MSC and/or increased bone matrix deposition in addition to decreased bone destruction resulted from attenuation of osteosarcoma tumor growth and invasion in bone. Have the authors examined the effects of PKA-induced MSC conditional medium and calreticulin on osteoblast differentiation and bone matrix deposition?

2) Although the authors have shown that PKA-induced MSC conditional medium (suppl Figure 2) and recombinant calreticulin (Figure 5A) decreased tumor volume in bone, no evidence has been provided that the two treatments inhibit tumor cell proliferation and survival in vivo by immunohistochemistry detection of Ki-67 and cleaved-caspase 3.

3) The results presented for the Calreticulin-CD47 signaling axis dissected in Figure 5 appear incomplete. The metadata in panel F suggests high expression of Calr coupled with low expression of CD47 improves survival for patients with all types of cancer. The only data the authors present to evaluate whether the levels of CD47 affect its interaction with Calreticulin are shown on panel E, but in U2OS cells, low levels of CD47 prevent Calreticulin's tumor suppressor function. Since this suggests that the Calr-CD47 interaction differs between osteosarcoma cells and "all cancer" it would help if this were mentioned more explicitly in the manuscript. More importantly, this argument can be strengthened with the addition of further evidence, such as an MTT assay that tests the effect of the presence of both Calr and CD47 on Osteosarcoma viability, as done for PCOLCE and APP in Figure 7 panel E.

4) Is there any data demonstrating that the siRNA knockdown of CD47 and PCOLCE in Figure 5E and Figure 7F was effective? Although the western blot data in those panels is convincing, it is difficult to accept without evidence that the intended protein levels were reduced by siRNA.

5) Some graphs indicate statistically significant differences, but the difference in means looks minimal. Examples include Figure 1E, Figure 3A BMD, Figure 3B, Figure 3C, Figure 3D, Figure 5B, and 5C. A possible solution would be to include source data for the graphs, and/or to adjust the plotted graphs to include individual points measured.

*Reviewer #1 (Recommendations for the authors):*

1) The increases in both trabecular and cortical bone mass in PKA-induced MSC conditional medium (Figure 3) and recombinant calreticulin (Figure 5A-5C) treated mice may be in part due to enhanced osteoblast differentiation from MSC and/or increased bone matrix deposition in addition to decreased bone destruction resulted from attenuation of osteosarcoma tumor growth and invasion in bone. Have the authors examined the effects of PKA-induced MSC conditional medium and calreticulin on osteoblast differentiation and bone matrix deposition?

2) Although the authors have shown that PKA-induced MSC conditional medium (suppl Figure 2) and recombinant calreticulin (Figure 5A) decreased tumor volume in bone, no evidence has been provided that the two treatments inhibit tumor cell proliferation and survival in vivo by immunohistochemistry detection of Ki-67 and cleaved-caspase 3.

*Reviewer #2 (Recommendations for the authors):*

In this manuscript, Li et al. expand their recently published work wherein they interrogate the proteomes of mesenchymal stem cell (MSC) culture media following stimulation with signaling pathways that exacerbate cancerous states, such as WNT and PI3K, to identify tumor suppressor proteins. They found that activation of MSCs with PKA also results in conditioned media capable of reducing the viability and motility of osteosarcoma cell lines in vitro. Moreover, this conditioned media was synergistic with anti-cancer drugs, effectively reduced the size of osteosarcoma tumors implanted in the bones of mice, and restored the bone loss caused by these tumors. By comparing transcripts and proteins identified within PKA-induced MSC conditioned media with public cancer databases, they investigated whether several proteins high in osteosarcoma in humans might paradoxically function as tumor suppressors. They focused on CALRETICULIN-CD47, PCOLCE-APP, and CALRETICULIN-APP interactions to highlight their approach as a proof of concept that perhaps identifying the proteins secreted by cancer cells can offer novel approaches to target osteosarcomas. Although they present substantial work, the following issues might be worth addressing.

– The results presented for the Calreticulin-CD47 signaling axis dissected in Figure 5 appear incomplete. The metadata in panel F suggests high expression of Calr coupled with low expression of CD47 improves survival for patients with all types of cancer. The only data the authors present to evaluate whether the levels of CD47 affect its interaction with Calreticulin are shown on panel E, but in U2OS cells, low levels of CD47 prevent Calreticulin's tumor suppressor function. Since this suggests that the Calr-CD47 interaction differs between osteosarcoma cells and "all cancer" it would help if this were mentioned more explicitly in the manuscript. More importantly, this argument can be strengthened with the addition of further evidence, such as an MTT assay that tests the effect of the presence of both Calr and CD47 on Osteosarcoma viability, as done for PCOLCE and APP in Figure 7 panel E.

– Is there any data demonstrating that siRNA knockdown of CD47 and PCOLCE in Figure 5E and Figure 7F was effective? Although the western blot data in those panels is convincing, it is difficult to accept without evidence that the intended protein levels were reduced by siRNA.

– There are enough differences between CW-CM combined with the different chemotherapeutic agents that should probably be described better in the Results section. The authors mention that "Importantly, the simultaneous application of CW-MSC CM significantly lowered the effective concentrations of MTX, DOX, and CIS in both TT2 and U2OS cells." Also, what is the sample size for the data plotted? At what doses were statistically significant differences observed?

– Some graphs indicate statistically significant differences, but the difference in means look minimal. Examples include Figure 1E, Figure 3A BMD, Figure 3B, Figure 3C, Figure 3D, Figure 5B, and 5C. A possible solution would be to include source data for the graphs, and/or to adjust the plotted graphs to include individual points measured.

– Some of the writing can use further proofreading.

---

## [Author Response]

Essential revisions:1) The increases in both trabecular and cortical bone mass in PKA-induced MSC conditional medium (Figure 3) and recombinant calreticulin (Figure 5A-5C) treated mice may be in part due to enhanced osteoblast differentiation from MSC and/or increased bone matrix deposition in addition to decreased bone destruction resulted from attenuation of osteosarcoma tumor growth and invasion in bone. Have the authors examined the effects of PKA-induced MSC conditional medium and calreticulin on osteoblast differentiation and bone matrix deposition?

Thank you for the comment on osteoblast differentiation. In the revised manuscript, we tested the effects of PKA-induced MSC conditioned medium (CM) and calreticulin on the differentiation and mineralization of osteoblasts using MC3T3 osteoblast cells. The differentiation was evaluated by examining the expression levels of type I collagen, alkaline phosphatase, and osteocalcin by Western blotting. The matrix deposition was evaluated by Arizalin red staining in an osteogenic medium. We included the results in Figure 5—figure supplement 2 in the revision.

2) Although the authors have shown that PKA-induced MSC conditional medium (suppl Figure 2) and recombinant calreticulin (Figure 5A) decreased tumor volume in bone, no evidence has been provided that the two treatments inhibit tumor cell proliferation and survival in vivo by immunohistochemistry detection of Ki-67 and cleaved-caspase 3.

Thank you for the comment on the effect of PKA-induced MSC-conditioned medium and calreticulin on the expression of Ki-67 and cleaved Caspase 3 in tumor-invaded bone sections. As suggested, we conducted immunohistochemistry and detected these two proteins in the section of the proximal tibia. We included the results in Figure 3—figure supplement 3 and Figure 5—figure supplement 1 in the revision.

3) The results presented for the Calreticulin-CD47 signaling axis dissected in Figure 5 appear incomplete. The metadata in panel F suggests high expression of Calr coupled with low expression of CD47 improves survival for patients with all types of cancer. The only data the authors present to evaluate whether the levels of CD47 affect its interaction with Calreticulin are shown on panel E, but in U2OS cells, low levels of CD47 prevent Calreticulin's tumor suppressor function. Since this suggests that the Calr-CD47 interaction differs between osteosarcoma cells and "all cancer" it would help if this were mentioned more explicitly in the manuscript. More importantly, this argument can be strengthened with the addition of further evidence, such as an MTT assay that tests the effect of the presence of both Calr and CD47 on Osteosarcoma viability, as done for PCOLCE and APP in Figure 7 panel E.

As suggested, we evaluated the CALR-CD47 interactions using the same approach to evaluate the PCOLCE-APP interactions in Figure 7E. Silencing CD47 also downregulated MTT-based viability, whereas it significantly suppressed Calr-induced tumor inhibition in U2OS OS cells, TT2 OS cells, MDA-MB-231 breast cancer cells, and PANC1 pancreatic cells. We included the results in Figure 5—figure supplement 4 in the revision.

4) Is there any data demonstrating that the siRNA knockdown of CD47 and PCOLCE in Figure 5E and Figure 7F was effective? Although the western blot data in those panels is convincing, it is difficult to accept without evidence that the intended protein levels were reduced by siRNA.

We apologize for not showing the silencing effect by siRNA for CD47 in Figure 5E. We included the results in Figure 5—figure supplement 3 in the revision.

5) Some graphs indicate statistically significant differences, but the difference in means looks minimal. Examples include Figure 1E, Figure 3A BMD, Figure 3B, Figure 3C, Figure 3D, Figure 5B, and 5C. A possible solution would be to include source data for the graphs, and/or to adjust the plotted graphs to include individual points measured.

We appreciate the comment and suggestion. We included source data for the graphs in addition to the description of the mean and standard deviation. Particularly, individual data points were included in most of the plots.

Reviewer #1 (Recommendations for the authors):1) The increases in both trabecular and cortical bone mass in PKA-induced MSC conditional medium (Figure 3) and recombinant calreticulin (Figure 5A-5C) treated mice may be in part due to enhanced osteoblast differentiation from MSC and/or increased bone matrix deposition in addition to decreased bone destruction resulted from attenuation of osteosarcoma tumor growth and invasion in bone. Have the authors examined the effects of PKA-induced MSC conditional medium and calreticulin on osteoblast differentiation and bone matrix deposition?

Done

2) Although the authors have shown that PKA-induced MSC conditional medium (suppl Figure 2) and recombinant calreticulin (Figure 5A) decreased tumor volume in bone, no evidence has been provided that the two treatments inhibit tumor cell proliferation and survival in vivo by immunohistochemistry detection of Ki-67 and cleaved-caspase 3.

Done

Reviewer #2 (Recommendations for the authors):In this manuscript, Li et al. expand their recently published work wherein they interrogate the proteomes of mesenchymal stem cell (MSC) culture media following stimulation with signaling pathways that exacerbate cancerous states, such as WNT and PI3K, to identify tumor suppressor proteins. They found that activation of MSCs with PKA also results in conditioned media capable of reducing the viability and motility of osteosarcoma cell lines in vitro. Moreover, this conditioned media was synergistic with anti-cancer drugs, effectively reduced the size of osteosarcoma tumors implanted in the bones of mice, and restored the bone loss caused by these tumors. By comparing transcripts and proteins identified within PKA-induced MSC conditioned media with public cancer databases, they investigated whether several proteins high in osteosarcoma in humans might paradoxically function as tumor suppressors. They focused on CALRETICULIN-CD47, PCOLCE-APP, and CALRETICULIN-APP interactions to highlight their approach as a proof of concept that perhaps identifying the proteins secreted by cancer cells can offer novel approaches to target osteosarcomas. Although they present substantial work, the following issues might be worth addressing.– The results presented for the Calreticulin-CD47 signaling axis dissected in Figure 5 appear incomplete. The metadata in panel F suggests high expression of Calr coupled with low expression of CD47 improves survival for patients with all types of cancer. The only data the authors present to evaluate whether the levels of CD47 affect its interaction with Calreticulin are shown on panel E, but in U2OS cells, low levels of CD47 prevent Calreticulin's tumor suppressor function. Since this suggests that the Calr-CD47 interaction differs between osteosarcoma cells and "all cancer" it would help if this were mentioned more explicitly in the manuscript. More importantly, this argument can be strengthened with the addition of further evidence, such as an MTT assay that tests the effect of the presence of both Calr and CD47 on Osteosarcoma viability, as done for PCOLCE and APP in Figure 7 panel E.

Done

– Is there any data demonstrating that siRNA knockdown of CD47 and PCOLCE in Figure 5E and Figure 7F was effective? Although the western blot data in those panels is convincing, it is difficult to accept without evidence that the intended protein levels were reduced by siRNA.

Done

– There are enough differences between CW-CM combined with the different chemotherapeutic agents that should probably be described better in the Results section. The authors mention that "Importantly, the simultaneous application of CW-MSC CM significantly lowered the effective concentrations of MTX, DOX, and CIS in both TT2 and U2OS cells." Also, what is the sample size for the data plotted? At what doses were statistically significant differences observed?

Thank you for the comment. We included source data for the graphs in addition to the description of the mean and standard deviation. Particularly, individual data points were included in most of the plots.

– Some graphs indicate statistically significant differences, but the difference in means look minimal. Examples include Figure 1E, Figure 3A BMD, Figure 3B, Figure 3C, Figure 3D, Figure 5B, and 5C. A possible solution would be to include source data for the graphs, and/or to adjust the plotted graphs to include individual points measured.

Done

– Some of the writing can use further proofreading.

Thank you for the comment.